# YOLOv8-DBW: An Improved YOLOv8-Based Algorithm for Maize Leaf Diseases and Pests Detection

**DOI:** 10.3390/s25154529

**Published:** 2025-07-22

**Authors:** Xiang Gan, Shukun Cao, Jin Wang, Yu Wang, Xu Hou

**Affiliations:** College of Mechanical Engineering, University of Jinan, Jinan 250022, China; 19106418250@163.com (X.G.); 15275583170@163.com (J.W.); 17605613474@163.com (Y.W.); 13061425596@163.com (X.H.)

**Keywords:** maize diseases and insect pests, YOLOv8, lightweight, target detection

## Abstract

To solve the challenges of low detection accuracy of maize pests and diseases, complex detection models, and difficulty in deployment on mobile or embedded devices, an improved YOLOv8 algorithm was proposed. Based on the original YOLOv8n, the algorithm replaced the Conv module with the DSConv module in the backbone network, which reduced the backbone network parameters and computational load and improved the detection accuracy at the same time. Additionally, BiFPN was introduced to construct a bidirectional feature pyramid structure, which realized efficient information flow and fusion between different scale features and enhanced the feature fusion ability of the model. At the same time, the Wise-IoU loss function was combined to optimize the training process, which improved the convergence speed and regression accuracy of the loss function. The experimental results showed that the precision, recall, and mAP0.5 of the improved algorithm were improved by 1.4, 1.1, and 1.5%, respectively, compared with YOLOv8n, and the model parameters and computational costs were reduced by 6.6 and 7.3%, respectively. The experimental results demonstrate the effectiveness and superiority of the improved YOLOv8 algorithm, which provides an efficient, accurate, and easy-to-deploy solution for maize leaf pest detection.

## 1. Introduction

Maize, as a strategic crop with the attributes of food, feed, and industrial raw materials worldwide, plays a crucial role in maintaining food security and driving economic development [1,2,3]. However, frequent pest and disease outbreaks (such as pests eating leaves and fruits and viruses infecting plants) severely restrict maize yield and quality, becoming a bottleneck for the sustainable development of the industry [4]. The 2021 report of the Food and Agriculture Organization of the United Nations (FAO) shows that global crop production is reduced by 20% to 40% annually due to pests and diseases [5], highlighting the urgency and importance of pest and disease control.

In agricultural production, the early and precise identification and detection of maize diseases and pests are of great significance for ensuring food safety, improving the quality and efficiency of agricultural products, and preventing economic losses. It is a key factor for the sustainable development of agriculture [6]. The traditional manual visual inspection method relies on the experience of technicians to determine the type and severity of diseases and pests. However, its drawbacks are obvious: the differences in knowledge and practical experience among different personnel can lead to significant deviations in judgment, and it is highly subjective. At the same time, inspecting each plant in large areas of farmland one by one is time-consuming and labor-intensive, with extremely low efficiency. It is difficult to meet the demand for the efficient and accurate identification of diseases and pests in the process of agricultural scale-up and modernization, and it is impossible to achieve timely and precise prevention and control.

With the development of computers and machine learning technologies, machine vision technology based on neural networks has been widely applied in crop pest and disease detection due to its advantages, such as high recognition efficiency and short recognition time [7]. Early research on plant disease recognition mainly relied on traditional machine learning methods, following the technical paradigm of “feature engineering + classifier”: extracting lesion areas through image preprocessing, such as threshold segmentation and morphological operations; manually designing features such as HSV color features, GLCM texture features [8], LBP operator [9], and Hu invariant moments as shape features; and then combining them with classifiers such as SVM [10], random forest [11], or KNN [12] to build models. Although these methods are effective under specific experimental conditions, they have obvious shortcomings: they have low recognition accuracy; are prone to confusion with similar pest and disease features, especially when the early symptoms of pests and diseases are not obvious; the accuracy is even worse; the image data processing flow is complex; the recognition speed is slow; and it is easily affected by environmental factors such as light and occlusion. When monitoring large areas, it is difficult to output results in a timely manner, which may delay the prevention and control opportunities; in addition, the multi-level structure of the model makes training and deployment difficult, with high hardware requirements. Grassroots agricultural units are restricted by equipment and find it difficult to apply, seriously limiting their promotion and practical application value.

Since Hinton et al. first introduced the concept of deep learning [13], this technology has developed rapidly in the field of pest and disease detection. Deep learning, by constructing an end-to-end feature learning framework, has greatly reduced the workload of manual feature engineering and significantly improved the speed and accuracy of pest and disease recognition [14]. Its core technical process includes first building a large-scale training set through data preprocessing methods such as random cropping and standardization normalization; then, designing a multi-layer convolutional neural network (CNN) architecture to automatically extract hierarchical features, where the shallow network is responsible for capturing low-level visual features such as edges and textures, and the deep network is used to aggregate morphological features and semantic information of disease spots; finally, using the backpropagation algorithm combined with optimization mechanisms such as the cross-entropy loss function to drive the iterative update of network parameters and feature representation learning [15]. A large number of studies have shown that deep learning methods have significantly outperformed traditional machine learning techniques in terms of recognition accuracy.

Current research on crop pest and disease recognition based on deep learning mainly falls into two technical routes: two-stage detection algorithms represented by R-CNN [16], Fast R-CNN [17], and Faster R-CNN [18], which are characterized by achieving high-precision detection through “candidate region generation-precise localization and classification”. For instance, Fuentes et al. combined Faster R-CNN and ResNet to achieve real-time recognition of tomato pests and diseases [19]; Zhang et al. proposed MF3R-CNN, which achieved an mAP of 83.34% in real scenarios [20]; and Alruwaili M et al. proposed the RTF-RCNN model for tomato leaf disease detection, and its accuracy rate was 97.42%, which was higher than Alex net (96.32%) and the CNN model (92.21%) [21]. However, these algorithms have high computational complexity and strict hardware requirements, which restrict their application at the grassroots level.

Another category is the single-stage detection algorithms represented by SSD [22] and the YOLO series [23], which do not require the generation of candidate regions and have a significant advantage in detection speed [24]. Among them, the YOLO algorithm has become a research hotspot due to its balance of accuracy and real-time performance: Yang Li et al. designed the DAC-YOLOv4 model, achieving an average precision of 72.7% for strawberry powdery mildew detection [25]; Yang et al. proposed Maize-YOLO, which optimized detection performance by integrating modules such as CSPResNeXt-50 [26]; and Yang H et al. proposed the YOLO-SDW algorithm, which enhanced model adaptability by introducing SPD-Conv and the Wise-IoU V3 loss function [27]. However, existing models often increase in complexity in pursuit of accuracy or speed, leading to a decline in real-time performance and an increase in deployment costs. There is an urgent need for detection solutions that balance accuracy, speed, and lightness.

Ultralytics released the open-source target detection model YOLOv8 in January 2023, which has advanced the field of target detection research [28]. Chen Z et al. proposed the YOLOv8-ACCW model by integrating AKConv, the CA mechanism, the CARAFE module, and the Wise-IoU loss function to optimize feature extraction and detection accuracy. The results demonstrated that the model achieved an F1 score, mAP50, and mAP50-95 of 92.4%, 92.8%, and 73.8%, respectively, representing increases of 3.1%, 3.1%, and 5.6%, compared to the original algorithm, while reducing model parameters and computational cost by 6.6% and 8.5%, respectively [29]. Jiang T et al. developed the lightweight YOLOv8-GO model by introducing a global attention mechanism (GAM) before the SPPF layer and optimizing the basic convolution, bottleneck, and C2F module to enhance feature fusion and reduce computational complexity [30]. This model achieved an mAP50 of 88.4% with 9.1 GFLOPs and a running speed of 275.1 FPS, making it suitable for resource-constrained environments. Chen D et al. proposed the YOLOv8-MDN-Tiny model for small-scale passion fruit disease detection, which used an MFSO structure to expand small target feature pixels and an improved DyRep module for multi-scale feature fusion. With the NWD loss function and model compression techniques, this lightweight model outperformed YOLOv8s in small target detection accuracy [31].

In the field of crop pest and disease detection, although some progress has been made in previous studies, there is still a contradiction between model complexity and embedded deployment. To address this issue, this paper proposed a lightweight maize leaf pest and disease detection method based on the improved YOLOv8: the Distribution Shift Convolution (DSConv) module was adopted to replace the traditional Conv layer, reducing memory usage and computational cost through kernel decomposition. BiFPN was introduced to construct a bidirectional feature pyramid, enhancing the multi-scale feature fusion capability. The Wise-IoU loss function was used to optimize the bounding box regression, improving the detection accuracy of complex pests and diseases and achieving a coordinated optimization of detection performance and a lightweight model.

## 2. Materials and Methods

### 2.1. Data Collection

The dataset used in this study consisted of two parts: one part from Kaggle and the other part obtained through self-collection manually in the field. The Kaggle dataset provided rich and diverse images of maize leaf diseases and insect pests, covering a variety of pest types, including small spot disease, large spot disease, corn borer, etc. These images were taken under different environmental conditions, with diverse lighting, angles, and backgrounds, which helped the model learn robust feature representations. Self-collected datasets were supplemented with scene-specific pest images taken in real-world field environments, including maize leaves at different stages of growth and damage levels. Self-collected data were strictly labeled and filtered to ensure the quality and diversity of data. The dataset consisted of nine categories, including fall armyworm larva, yellow stem borer, yellow stem borer larva, grasshopper, gray leaf spot, corn rust, downy mildew, blight, and healthy (as shown in Figure 1).

### 2.2. Data Enhancement and Annotation

By combining these two datasets and using OpenCV (4.10.0.84) for data enhancement, including flip, rotation, saturation adjustment, and noise (as shown in Figure 2), we constructed a comprehensive and representative dataset for training and validating the target detection model. The dataset consisted of 12,440 images. LabelImg annotation software (1.8.6) was used to annotate maize leaf diseases and pests and to mark the positions of different types of lesions in the images. The dataset was then randomly divided into a training set and validation set at a ratio of 9:1 and formatted in YOLO format with dataset annotation file statistics and visualizations, as shown in Figure 3. From left to right and top to bottom, they respectively represent the data volume of the training set, the size and quantity of annotated bounding boxes, the position of the center point of the boxes relative to the entire image, and the height–width ratio of the targets in the image relative to the entire image.

### 2.3. Improved Method

#### 2.3.1. YOLOv8 Model

YOLOv8 continues the YOLO series single-stage target detection framework and adopts the backbone–neck–head three-level architecture design to achieve a breakthrough in the balance between accuracy and speed. The structure diagram and working principle are shown in the following Figure 4.

Input image preprocessing: The input image is first scaled to a fixed size preset by the model (e.g., 640 × 640 pixels) and usually normalized to a specific range (e.g., 0 to 1) to facilitate network training and convergence. To preserve the original aspect ratio of the image during scaling and to avoid distortion of the image content, Letterbox or Pad operations are often used.

Backbone network: The preprocessed image is sent to the backbone network, whose main function is to extract different levels of visual features from the input image. YOLOv8 ‘s backbone network adopts the CSPDarkNet structure and replaces the C3 module in YOLOv5 with the C2f module, becoming more lightweight. The C2f module reduces the computational effort by adding more layer-hopping connections and additional split operations while increasing gradient flow to improve convergence speed and effectiveness.

Neck network: The neck network uses the SPPF module to convert feature maps of different sizes into feature vectors of a consistent size. SPPF reduces computation and improves speed by connecting the three largest pooling layers step by step, compared to spatial pyramid pooling (SPP) structures. At the same time, in order to improve the recognition performance of the model, the PANet architecture is adopted in the neck part, which enhances the ability of the network to fuse different-scale target features and is used to propagate feature information and merge different levels of features.

Head network: Detection and classification are separated by decoupling the head structure. In addition, YOLOv8 switches from the anchor-based method to the anchor-free method, which significantly reduces computation time and improves speed without degrading accuracy. In terms of loss function calculation, YOLOv8 uses VFL Loss as classification loss (BCE Loss is used in actual training), while DFL Loss and CIOU Loss are used as regression losses. YOLOv8 improves the label assignment strategy, abandoning the previous IoU assignment or one-sided proportional assignment, and instead adopts the task-aligned assignment strategy for the allocation of positive and negative samples.

#### 2.3.2. DSConv Module

DSConv is an efficient convolutional operator designed to reduce the memory usage and computational complexity of neural networks by quantization while maintaining high accuracy. DSConv decomposes the conventional convolution kernel into a variable quantization kernel (VQK) and a distribution offset. Only integer values are stored in VQK to reduce memory usage, while the distribution offset preserves the output characteristics of the original convolution. In addition, DSConv quantizes activation values using the block floating point (BFP) method, with each block sharing an exponent, thereby reducing precision loss. This method significantly improves the calculation speed by replacing floating-point operations with integer operations.

As shown in Figure 5, DSConv is a method of modeling convolution behavior using quantization and distribution shifts, where the symbol ◎ denotes the Hadamard operator. The method consists of two parts: variable quantized kernel (VQK) and distributed shift. The VQK only holds integer values of variable bit length, which is part of the quantized component of DSConv, which makes multiplication faster and storage more efficient; the purpose of the distribution shift is to shift the distribution of VQK to mimic the distribution of the original convolution kernel. “Shift” refers to scaling and biasing operations, which are implemented with two tensors: the kernel distribution shifter (KDS), which shifts the distribution in each block of VQK, and the channel distribution shifter (CDS), which shifts the distribution in each channel. With DSConv, convolution kernels can be reduced to a fraction of their original size, enabling faster, more memory-efficient computations.

#### 2.3.3. BiFPN Module

Feature pyramid networks (FPNs) are often used to deal with multi-scale target detection problems in target detection tasks. However, traditional FPNs may be inadequate in dealing with complex features. To solve this problem, YOLOv8 adopts the PANet network based on FPN improvement in the feature fusion part. PANet effectively promotes information flow through the top-down one-way fusion and bottom-up secondary fusion mechanism of FPNs. Compared with FPNs, PANet’s bottom-up feature propagation method is more efficient and can achieve a better semantic segmentation effect with fewer computational resources. However, PANet’s two-way fusion mechanism is relatively simple.

To improve the performance of the model, BiFPN (bi-directional feature pyramid network) emerged as an improved scheme and was successfully applied to the neck part of YOLOv8 to achieve finer feature fusion (as shown in Figure 6). BiFPN enables efficient information flow and fusion between features at different scales by constructing a bidirectional feature pyramid from top to bottom and bottom to top. Specifically, BiFPN consists of multiple levels of feature pyramids, each level fusing feature maps from different scales. This two-way fusion mechanism ensures that the feature map at each level contains rich multi-scale information, thus improving the detection accuracy of the model for different-sized targets. In addition, BiFPN also uses a weighted feature fusion strategy, assigning different weights to different feature maps to enhance the influence of important features and suppress the interference of unrelated features. In YOLOv8, BiFPN is integrated into the neck of the model, working in conjunction with the backbone and head to significantly improve the model’s feature extraction capabilities and target detection performance. Experimental results show that BiFPN can effectively improve the detection accuracy and robustness of the model while keeping the model lightweight, especially when dealing with multi-scale targets.

#### 2.3.4. Wise-IoU

IoU is a common evaluation metric for object detection models. It measures the overlap between the predicted bounding box and the true bounding box. However, IoU evaluation results can be biased when dealing with objects of different sizes. The loss function used in the YOLOV8 algorithm is CIoU. CIoU is a common evaluation index and loss function in the field of object detection, which is used to measure the similarity between predicted frames and real frames while ignoring classification information. The classification of complex and diverse pest characteristics is particularly important in the detection of maize leaf diseases and insect pests. Although CIoU adopts a monotonic focus mechanism to improve target detection accuracy, it does not fully balance the gradient contributions of difficult and easy samples. When the training set contains low-quality examples, model performance may suffer. Its formula is as follows:(1)$$\begin{array}{c} CIoU=IoU-\frac{\rho^{2}\left(b_{,}b^{gt}\right)}{c^{2}}-\alpha v \end{array}$$

As can be seen from the above equation, ρ represents the Euclidean distance between the center points of the prediction frame and the real frame, and b represents the center points of the prediction. b^gt^ represents the center point of the true box, and c represents the linear distance between the predicted box and the true box. v is a parameter used to measure aspect ratio consistency, α is a weighting function used to adjust distance penalty, and v and α are expressed as follows:(2)$$\begin{array}{c} v=\frac{4}{\pi }{\left(\arctan\frac{\omega^{gt}}{h^{gt}}-arctan\frac{w}{h}\right)}^{2} \end{array}$$(3)$$\begin{array}{c} \alpha =\frac{v}{\left(1-IoU\right)+v} \end{array}$$

In this study, a Wise-IoU loss function with a dynamic nonmonotonic focusing mechanism was introduced to balance the samples. It evaluated the quality of anchor frames by introducing the concept of outlier degree and designed a gradient gain allocation strategy to balance the gradient contribution of high-quality and low-quality anchor frames. Specifically, Wise-IoU reduced the competitiveness of high-quality anchor boxes while reducing the harmful gradients produced by low-quality anchor boxes, making the model more focused on moderate-quality anchor boxes. The core formula for Wise-IoU is as follows:(4)LWIoU=rRWIoULIoU(5)r=βδaβ−α(6)β=LIoU*LIoU¯(7)RWIoU=ⅇxpx−xy2+y−ygt2Cw2+Cn2*
where L_IoU_ ∈ [0, 1] denotes the IoU loss, which weakens the penalty for high-quality anchor boxes and increases the focus on the distance between centers when the overlap between anchor points and prediction boxes is high; and R_WIoU_ ∈ [1, exp] denotes the penalty term in Wise-IoU, which increases the loss of average-quality anchor boxes. Superscript ∗ means not participating in backpropagation, thereby effectively preventing gradients that might prevent the network model from converging. LIoU¯ acts as a normalization factor and represents an incremental moving average. β represents the outlier score, where lower values indicate higher anchor box quality and assign smaller gradient lifts.

Because the characteristics of maize leaf diseases and insect pests are different in size, small characteristics (such as early disease spots of some diseases) and large characteristics (such as some obvious pests) coexist. The traditional IoU loss function has limitations in dealing with this kind of multi-scale target detection. Therefore, the Wise-IoU loss function was chosen for this study. Wise-IoU can more accurately measure the degree of bounding box overlap of different-sized targets and reduce the negative impact of low-quality examples on model training by dynamically adjusting gradient contributions, thus improving the detection performance of the model for multi-scale targets.

#### 2.3.5. YOLOv8-DBW Network Model Architecture

In order to obtain a new detection model for maize leaf diseases and insect pests, YOLOv8-DBW (as shown in Figure 7) was proposed based on YOLOv8n of the YOLOv8 model series (including YOLOv8n, s, m, l, and x, model sizes from small to large). By integrating DSConv into backbone and neck networks instead of the Conv module, BiFPN was used to optimize the multiscale feature fusion method to achieve effective bidirectional cross-scale connection and weighted feature fusion. Finally, Wise-IoU was used as a detector evaluation index instead of IoU to provide more accurate evaluation results.

## 3. Results

### 3.1. Experimental Environment and Parameter Settings

To ensure the effectiveness of deep learning model training, this study built a targeted experimental platform. The experimental environment was based on the Windows 11 operating system. The specific system hardware configuration and software development environment are shown in Table 1.

Based on the hardware configuration and experimental requirements, this study optimized the parameters, aiming to balance the computing resource consumption and model performance, improve the training efficiency, and accelerate the model convergence. The setup system ensured that the experiment was carried out stably under unified conditions. The specific parameter configuration is detailed in Table 2.

### 3.2. Evaluation Index

Precision (P), recall (R), F1 score, mean precision (mAP), frames per second (FPS), memory usage (MB), number of parameters (Params), and giga floating-point operations per second (GFLOP) were used in this study. F1 score and mAP served as core indicators, calculated as the harmonic average of the precision rate and recall rate and the precision average under different recall rates, respectively. These metrics measured the accuracy of model recognition, with mAP50 representing the average accuracy of the model when the IoU threshold was 0.5. FPS and MB respectively reflected the running speed and resource usage of the model. The number of parameters (Params) and GFLOP reflected the size and computational complexity of the model, respectively. The smaller the Params, the lighter the model was. The lower the GFLOP, the lower the computational requirements. The two cooperated to reflect the applicability of the model on resource-constrained devices.

Precision is a measure of the proportion of positive samples predicted by the model, calculated as follows:(8)$$\begin{array}{c} P=\frac{TP}{TP+FP} \end{array}$$

Recall measures the proportion of positive samples that the model correctly predicts, calculated as follows:(9)$$\begin{array}{c} R=\frac{TP}{TP+FN} \end{array}$$
where TP is the number of correctly predicted targets, FP is the number of incorrectly predicted targets, and FN is the number of incorrectly predicted targets.(10)$$\begin{array}{c} AP=\int_{0}^{1}P\left(R\right)d\left(R\right) \end{array}$$

AP is the area of the curve enclosed by P and R, indicating the average accuracy. The mAP value is obtained by averaging the AP values of each category, and the larger the mAP value, the better the performance of the model. The calculation formula is as follows:(11)$$\begin{array}{c} mAP=\frac{\sum_{i=1}^{n}AP_{i}}{n} \end{array}$$

The F1 score is the harmonic average of precision and recall and is often used to evaluate the overall performance of the model. It is calculated as follows:(12)$$\begin{array}{c} F1=2\frac{P\times R}{P+R} \end{array}$$

### 3.3. DSConv Module’s Impact on Network Performance

To explore the influence of the DSConv module on network performance, ablation experiments were designed shown in Table 3. The experimental settings were as follows: DSConv-backbone meant that only DSConv modules were integrated into the backbone network, replacing Conv modules in them while keeping the neck network structure unchanged; DSConv-neck was the opposite, only replacing Conv modules with DSConv modules in the neck network, and the backbone network remained unchanged. Further, DSConv-all covered both cases above, i.e., replacing Conv modules with DSConv in both backbone and neck networks. By comparing the experimental results of these three configurations, the aim was to systematically evaluate the contribution of DSConv modules to overall performance at different network locations.

DSConv-all meant replacing all Conv modules with DSConv modules in the backbone network and neck network of YOLOv8n. The DSConv module was introduced into the YOLOv8n backbone network to provide an efficient convolution mechanism. DSConv can flexibly adapt to different feature distributions by dynamically adjusting the weights of convolution kernels, effectively reducing memory usage and computational complexity while maintaining high accuracy. This module adopted a distributed offset and variable quantization kernel strategy, reduced the memory footprint, and sped up computation, especially suitable for resource-constrained environments. DSConv also introduced an asymmetric convolution kernel design, which enhanced the detection ability of the model for multi-scale targets by decomposing the convolution kernel into two branches and using different convolution kernel sizes for feature extraction. This method broke through the limitation of traditional convolution and enabled the model to deal with targets of different scales more efficiently. Experimental results showed that after replacing the Conv module with the DSConv module in YOLOv8n, the detection accuracy and adaptability to multi-scale targets were significantly improved while maintaining high efficiency.

### 3.4. Ablation Experiments

To verify the accuracy of the proposed improved algorithm, ablation tests were used for comparison, and the results are shown in Table 4 below, where √ represents the application of the module, and × represents that the module is not applied.

It can be seen from the table that the improved algorithm adopted a more efficient network structure to enhance the YOLOv8n network structure, improved accuracy, and reduced model parameters and computational complexity. This also proved that the DSConv module did not reduce the accuracy of the algorithm but reduced the model parameters and computational load. BiFPN was introduced to enhance the YOLOv8 algorithm to improve the adaptability of the model to targets of different scales. The Wise-IoU boundary loss function improved the evaluation ability of detectors and played a vital role in detecting multiple pests. Combining these improvements with the YOLOv8n algorithm, the model volume was minimized, the parameters were reduced to only 2.8 M, and the computational complexities were reduced to 7.6 G, which were 6.6% and 7.3%, respectively. This effectively reduced the difficulty and cost of deploying models on a mobile device and significantly improved accuracy while meeting real-time requirements.

### 3.5. Detection Comparison of Different Algorithms

To fully evaluate the performance of YOLOv8-DBW, this study used the maize leaf disease dataset to carefully compare it with seven mainstream target detection algorithms, including Faster R-CNN, SSD, YOLOv5s, YOLOv6n, YOLOv7Tiny, YOLOv8n, YOLOv9s, and YOLO11n shown in Table 5. Experimental results showed that YOLOv8-DBW had significant advantages in key performance indicators.

YOLOv8-DBW achieved a 16.8% improvement in mAP50 metrics compared to the two-stage detection algorithm Faster R-CNN and a 19.2% improvement compared to the single-stage detection algorithm SSD. This indicates that YOLOv8-DBW has remarkable superiority in detection accuracy. At the same time, YOLOv8-DBW also demonstrated significant improvements in accuracy (P) and recall (R), further highlighting its effectiveness and reliability in target detection tasks. YOLOv8-DBW also showed excellent performance when compared to other YOLO models. YOLOv8-DBW achieved 5.2%, 4.6%, 1.4%, 1.5%, and 0.4% improvements in mAP50 compared to YOLOv5s, YOLOv6n, YOLOv7Tiny, YOLOv8n, and YOLOv9s, respectively. Meanwhile, it achieved a performance close to that of the latest YOLO11n. These results reflect not only YOLOv8-DBW’s progress in algorithm optimization but also its ability to further improve detection accuracy while maintaining the consistently high speed of the YOLO series. It is worth noting that YOLOv8-DBW also showed obvious advantages in terms of model parameter quantity. YOLOv8-DBW has the smallest model size among all compared algorithms, which means it requires the least storage space and memory resources when deployed. This is a critical advantage for resource-constrained edge computing devices and mobile applications because it significantly reduces hardware costs and energy consumption.

In summary, the YOLOv8-DBW algorithm stands out among many target detection algorithms due to its significant advantages in detection accuracy and model volume. These features make YOLOv8-DBW the preferred solution for practical applications, not only meeting high standards of accuracy and speed but also achieving an optimal balance between deployment costs and computational resource consumption, demonstrating excellent overall performance and broad application prospects.

### 3.6. Application of Experimental Results

To illustrate the performance improvement of YOLOV8-DBW over the baseline algorithm YOLOv8n, we compared their detection results on the same set of images. The test results are shown in Figure 8, where the test results of the baseline algorithm are displayed on the left, and YOLOV8-DBW test results are displayed on the right. Compared with the baseline algorithm, the improved algorithm could identify a variety of maize diseases and insect pests more accurately, resulting in higher detection accuracy. The experimental data showed that compared with the baseline algorithm, the YOLOV8-DBW algorithm proposed in this paper showed significant advantages. In the task of maize leaf disease and pest detection, the algorithm could not only accurately locate the disease target but also classify it with extremely high recognition accuracy. Even in the case of complex background or unclear disease and pest features, it still maintained excellent detection performance, which fully verifies its excellent robustness and detection accuracy.

To further verify the detection performance of the model for maize leaf disease and pests, Table 6 compares the performance of YOLOv8n and the improved YOLOV8-DBW model in detecting maize leaf disease and pests. It can be seen from the overall performance that the improved model performed better than the baseline model in detecting various diseases and pests and had more practical application value.

As shown in Figure 9, this study compared the precision-recall (PR) curves of the traditional YOLOv8n and YOLOv8-DBW models. The quantitative analysis results showed that the YOLOv8-DBW model showed a significant performance improvement in the detection of various disease targets in the detection task of maize leaf diseases and insect pests. Compared with the original model, the mAP50 was increased from 87.5% to 89.0%, an increase of 1.5%, which fully verifies the effectiveness of the improved algorithm in optimizing the target detection accuracy.

Figure 10 is the confusion matrix of the YOLOv8n and YOLOv8-DBW models. In the visualization of the confusion matrix, the deeper the gray value of the main diagonal color block, the higher the correct detection rate of the model for the corresponding category. The deeper the gray value of the non-main diagonal color block, the more likely it was to reflect that the model was confused in the detection results of the two types of targets corresponding to the horizontal (predicted category) and vertical (actual category) categories.

## 4. Conclusions

An improved target detection model, YOLOv8-DBW, was proposed to detect maize leaf diseases and insect pests. Several enhancement techniques were introduced, including DSConv, BiFPN, and Wise-IoU, where DSConv replaced traditional convolution to significantly reduce memory usage and speed up computation while basically maintaining model accuracy. Secondly, BiFPN was introduced to construct a bidirectional feature pyramid structure, which realized efficient information flow and fusion among different-scale features and enhanced the feature fusion ability of the model. Finally, the Wise-IoU boundary loss function was used instead of CIoU in YOLOv8 to enhance the boundary box regression performance of the network and improve the detection of complex diseases and pests. The improved YOLOv8-DBW showed significant advantages in model accuracy, efficiency, and adaptability. Experimental results showed that YOLOv8-DBW achieved significant improvements in P, R, and mAP50 indices, and also reduced parameters and computational complexity compared with existing advanced target detection algorithms, achieving a balance between minimizing model volume and maximizing detection accuracy. These improvements not only greatly reduce the complexity and cost of model deployment on mobile devices but also significantly improve the detection efficiency so that it can better adapt to the real-time requirements in actual application scenarios.

Overall, YOLOv8-DBW provides a superior solution for maize leaf pest detection due to its high performance and wide applicability. Future work will focus on further optimizing the model structure to improve its robustness to extreme conditions and rare pest types while exploring its potential for application in other agricultural fields.

## Figures and Tables

**Figure 1 sensors-25-04529-f001:**
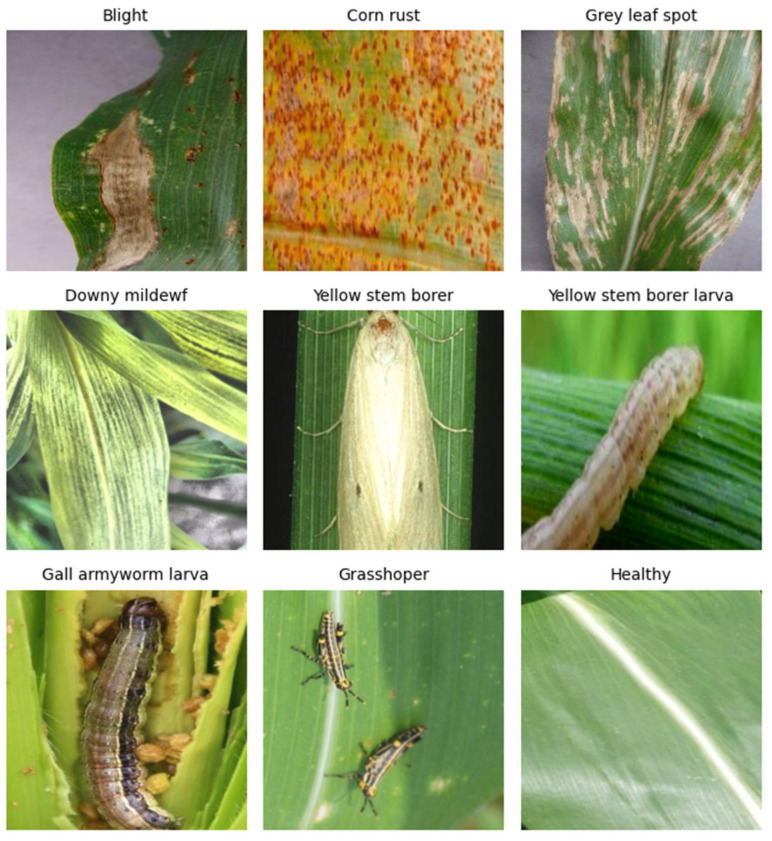
Sample images of maize leaf diseases and pests.

**Figure 2 sensors-25-04529-f002:**
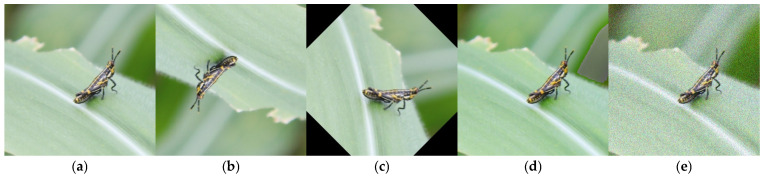
Image enhancement methods. (**a**) Original image; (**b**) image after flipping; (**c**) image after rotation; (**d**) image after saturation adjustment; (**e**) image after noise processing.

**Figure 3 sensors-25-04529-f003:**
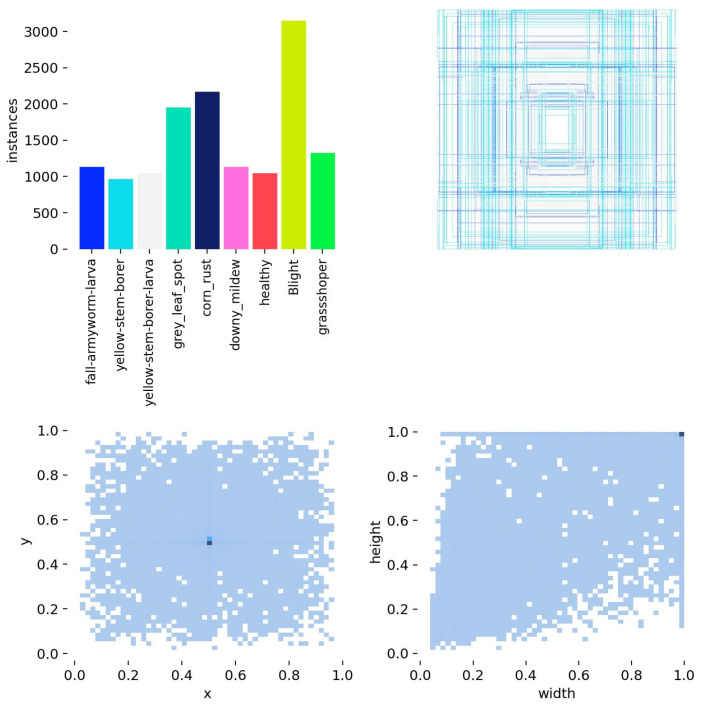
Maize leaf diseases and pests’ dataset annotation file statistics.

**Figure 4 sensors-25-04529-f004:**
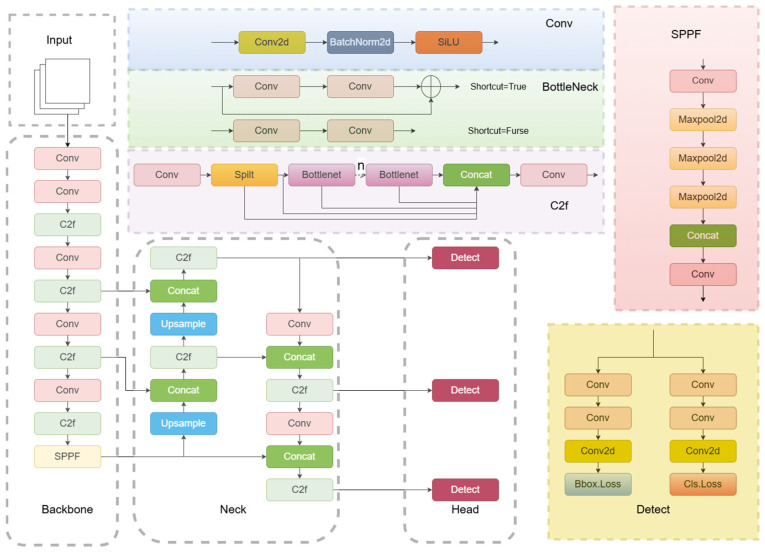
YOLOv8 network architecture.

**Figure 5 sensors-25-04529-f005:**
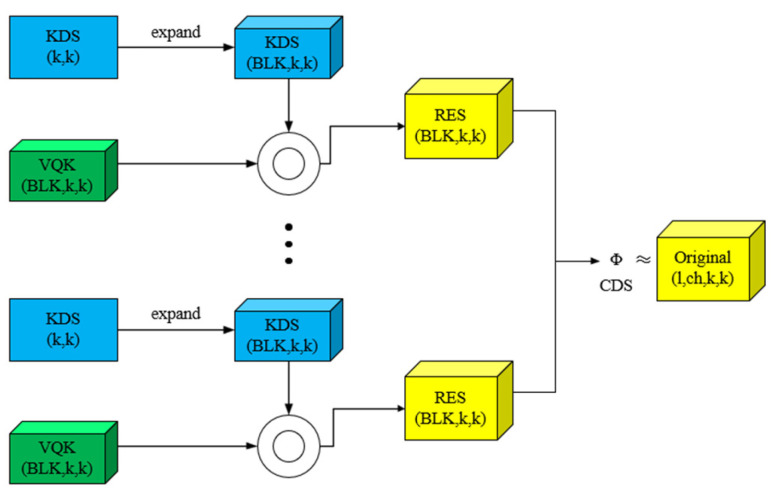
General idea of DSConv.

**Figure 6 sensors-25-04529-f006:**
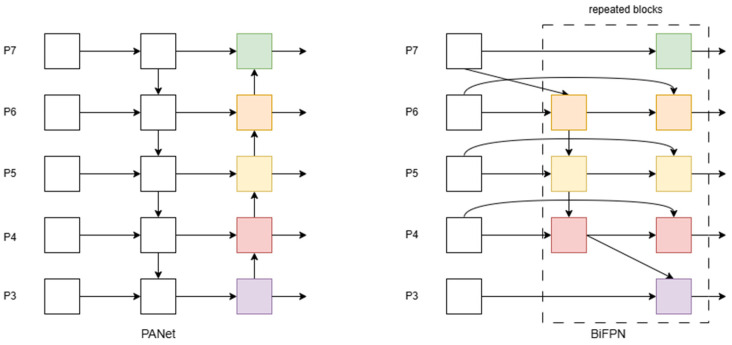
Architecture diagrams of PANet and BiFPN.

**Figure 7 sensors-25-04529-f007:**
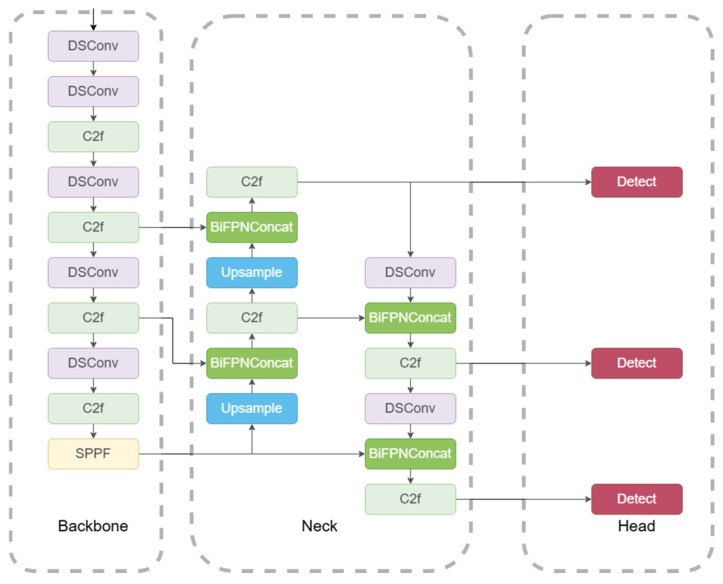
YOLOv8-DBW network architecture.

**Figure 8 sensors-25-04529-f008:**
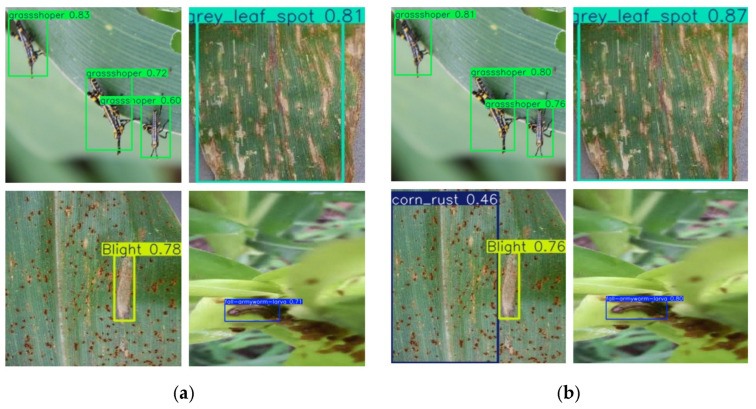
Comparison of detection effects. (**a**) YOLOv8n; (**b**)YOLOv8-DBW.

**Figure 9 sensors-25-04529-f009:**
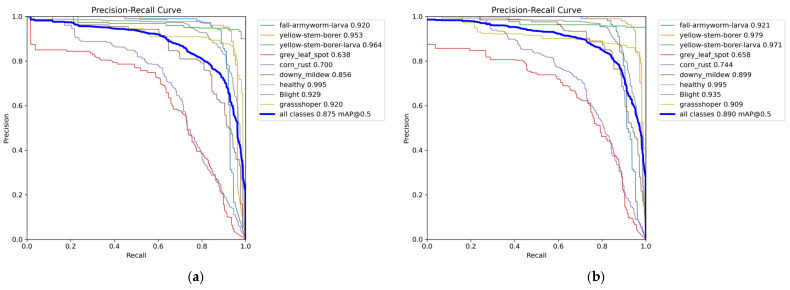
Precision–recall diagram. (**a**) YOLOv8n; (**b**) YOLOv8-DBW.

**Figure 10 sensors-25-04529-f010:**
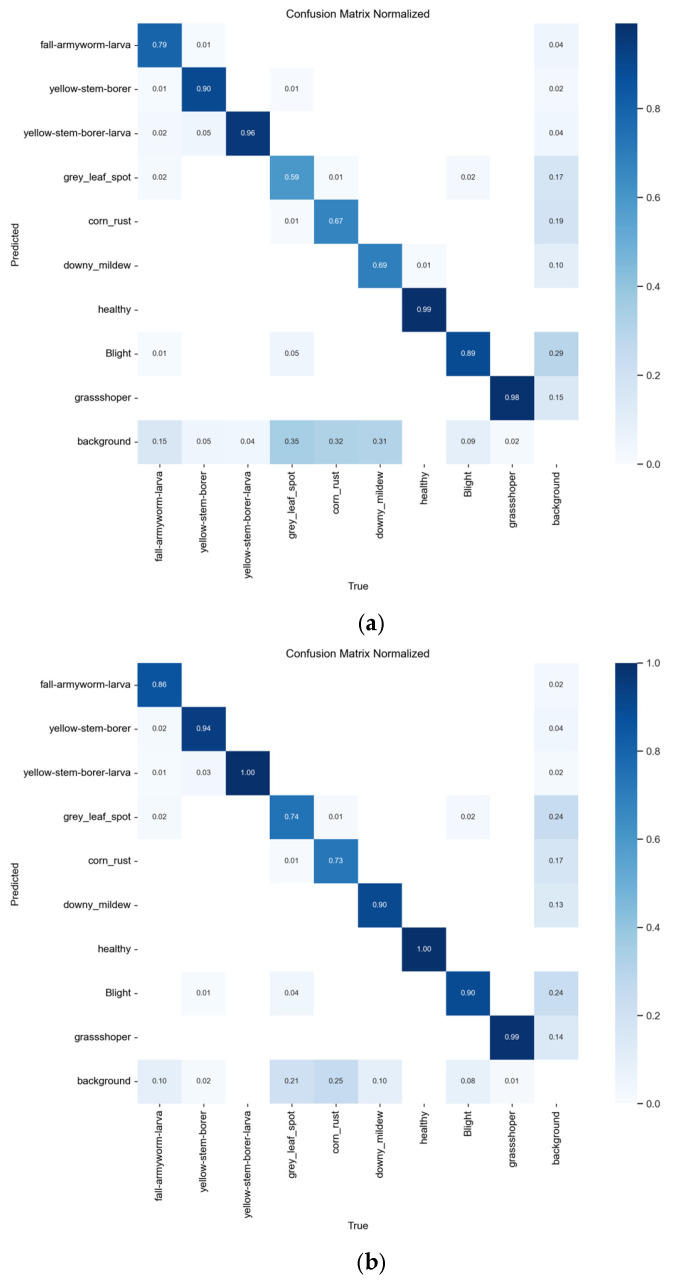
Confusion matrices. (**a**) YOLOv8n; (**b**) YOLOv8-DBW.

**Table 1 sensors-25-04529-t001:** Experimental environment configuration.

Software/Hardware Configuration	Information
CPU	Intel(R)Core (TM)i7-13650HX
GPU	NIVIDA RTX 4060
Memory	16 GB
Operating system	Windows 11
System environment	Pycharm
Python version	3.8.20
Pytorch version	2.2.1
CUDA version	11.8

**Table 2 sensors-25-04529-t002:** Experimental parameter settings.

Parameter Types	Parameter Settings
Image size	640 × 640
Initial learning rate	0.01
Batch	16
Epochs	100
Patience	50
Optimizer	SGD

**Table 3 sensors-25-04529-t003:** Comparison results of different positions of DSConv.

Algorithm	P	R	F1	mAP50
YOLOv8n	0.850	0.850	0.847	0.875
DSConv-backbone	0.856	0.849	0.849	0.880
DSConv-neck	0.843	0.864	0.851	0.878
DSConv-all	0.860	0.859	0.857	0.884

**Table 4 sensors-25-04529-t004:** Ablation experiment results.

Module				P	R	F1	mAP50	GFLOPs	Params
	DSConv	BiFPN	Wise-IoU						
YOLOv8n	×	×	×	0.850	0.850	0.847	0.875	8.2	3.0
	√	×	×	0.860	0.859	0.857	0.884	6.9	2.7
	×	√	×	0.845	0.858	0.850	0.883	8.1	3.0
	×	×	√	0.875	0.850	0.859	0.884	8.2	3.0
	√	√	×	0.870	0.837	0.850	0.881	7.6	2.8
	√	×	√	0.864	0.845	0.850	0.888	6.9	2.7
	×	√	√	0.866	0.837	0.847	0.879	8.1	3.0
	√	√	√	0.864	0.861	0.860	0.890	7.6	2.8

**Table 5 sensors-25-04529-t005:** Comparison results of different algorithms.

Model	P	R	mAP50	GFLOPs	FPS	MB	Params
Faster R-CNN	0.624	0.763	0.722	251.4	13	320	114.2
SSD	0.758	0.633	0.698	96.7	52	83	89.1
YOLOv5s	0.823	0.798	0.838	16.8	80	18.5	8.1
YOLOv6n	0.817	0.806	0.844	13.0	105	6.8	9.2
YOLOv7Tiny	0.844	0.837	0.876	13.5	68	6.0	6.6
YOLOv8n	0.850	0.850	0.875	8.2	286	6.0	3.0
YOLOv9s	0.855	0.852	0.886	26.7	273	5.8	7.2
YOLO11n	0.864	0.842	0.889	6.6	257	5.2	2.6
YOLOv8-DBW	0.864	0.861	0.890	7.6	282	6.1	2.8

**Table 6 sensors-25-04529-t006:** Comparison of model detection performance.

Category	YOLOv8n			YOLOv8-DBW		
	P	R	mAP50	P	R	mAP50
Fall armyworm larva	0.967	0.840	0.920	0.970	0.864	0.921
Yellow stem borer	0.907	0.935	0.953	0.920	0.954	0.979
Yellow stem borer larva	0.899	0.991	0.964	0.943	1.000	0.971
Grasshopper	0.796	0.963	0.920	0.814	0.975	0.909
Grey leaf spot	0.695	0.610	0.638	0.687	0.636	0.657
Corn rust	0.779	0.608	0.705	0.778	0.588	0.744
Downy mildew	0.777	0.809	0.856	0.827	0.846	0.899
Blight	0.841	0.895	0.929	0.861	0.890	0.935
Healthy	0.994	1.000	0.995	0.995	1.000	0.995
All	0.850	0.850	0.875	0.864	0.861	0.890

## Data Availability

The source code and public datasets in the present study may be available from the corresponding author upon request.

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
