# Peer review of "YOLOv8-DBW: An Improved YOLOv8-Based Algorithm for Maize Leaf Diseases and Pests Detection"

_sensors, 2025, doi:10.3390/s25154529_

Round 1
Reviewer 1 Report
Comments and Suggestions for Authors
This research proposes an improved YOLOv8-DBW algorithm, based on DSConv module, BiFPN structure and Wise - IoU loss function for the detection of corn pests and diseases, Overall, the article is good. However, there are still some problems that need to be revised.
- The paper repeatedly highlights the algorithm's deployment advantages, such as those stated in the end of the abstract "easy-to-deploy"and “This effectively reduces the difficulty and cost of deploying models on a Mobile device, and significantly improves accuracy while meeting real-time requirements” in line 391 but it is not mentioned the supporting data throughout the text.
- The description of the dataset used in this studyis unclear. One is From Kaggle, how can the readers find it; the other is self-collected datasets, how did you get it?where?when? Please supplement relevant information about data sources.
- Some figures are unclear, making it impossible to see the numbers on them, for example, Figure 10or some figures lack sufficient information, making it impossible to understand the meaning conveyed by them for example, Figure 3.
- There are duplicate entries in the references, or the reference links are incorrect.
Reviewer 2 Report
Comments and Suggestions for Authors
In order to solve the challenges of low detection accuracy of corn pests and dis-9 eases, complex detection models, and difficulty in deployment on mobile or embedded 10 devices, an improved YOLOv8 algorithm is proposed. However, there are still the following problems:
- The dataset consists of two parts. How many images are in the custom-created dataset? Additionally, many data augmentation methods were used. How much raw data existed before augmentation?
2.Integrating DSConv into YOLOv8 is a common practice. What is the innovation in this paper for doing so?
3.Integrating Wise-IoU into YOLOv8 is a common practice. What is the innovation in this paper for doing so?
4.BiFPN is also an existing method; merely using it offers relatively weak innovation.
5.Why doesn't the experimental section compare the results with YOLOv11?
6.Overall, this paper leans more towards engineering application and has weaker theoretical innovation. It needs to propose more novel viewpoints.
Reviewer 3 Report
Comments and Suggestions for Authors
The authors conducted a very valuable research. This paper appears to be well-organized and to be suitable for publication in the journal. I have some suggestions to need to be edited before publication.
@ Most figure captions are too brief. Figures should be understood independently of the text.
@ Simplification:
line 9 : "In order to" → "To"
line 17 : "optimize further" → "to optimize“
line 34 : “each year” → “annually”
line 412 : “of all the algorithms involved in the comparison”→ “among all compared algorithms
line 463 : “A variety of enhancement algorithms” → “several enhancement techniques”
@ Replacement with academic expressions and more appropriate expression
line 9 : "Corn" → "Maize“
line 13 : "Calculation amount" → "computational load"
line 14 : "Secondly" → "Additionally"
line 22 : "prove" → "demonstrate"
line 30 : “attacks”→ “outbreaks” 
line 36 : “self-collection” → “collected manually in the field”
line 110 and 403 : “showed” → “demonstrates”
line 431 : “obtain higher detection accuracy” → “resulting in...”
line 436 : “fuzzy characteristics” → “unclear features”
line 476 : “difficulty” → “complexity”
line 480 : “we plan to…” →“Future work will focus on…”
Round 2
Reviewer 2 Report
Comments and Suggestions for Authors
accept